# Effects of the Implementation of an Intervention Based on Falls Education Programmes on an Older Adult Population Practising Pilates–A Pilot Study

**DOI:** 10.3390/ijerph20021246

**Published:** 2023-01-10

**Authors:** María del Carmen Campos-Mesa, Marta Rosendo, Kevin Morton, Óscar DelCastillo-Andrés

**Affiliations:** 1Physical Education and Sport Department, University of Seville, 41013 Seville, Spain; 2School of Sport and Health Sciences, University of Brighton, Eastbourne BN20, UK

**Keywords:** adults, adapted utilitarian judo, safe fall-safe schools, Pilates method, quality of life

## Abstract

Age brings consequent physical deterioration of body balance, strength, flexibility and agility. It has been demonstrated that daily physical activity (PA), managed by professionals, is fundamental to ageing with increased quality and to reducing the number of falls, which are a consequence of factors highlighted above. This has been most evident during the COVID-19 pandemic. The aim of this study is to determine and analyse the effects of a multidisciplinary intervention based on the Safe Fall, Safe Schools, adapted utilitarian judo (JUA) and Pilates programmes in a population of older people. After an intervention of 60 min, 2 days a week for 12 weeks, the changes produced in variables such as quality of life, balance, lower body strength, flexibility and agility were analysed. A *p*-value ≤ 0.05 is accepted. The data show that the intervention can help to improve quality of life, especially two of its dimensions: pain (increases 12 points) and health transition (increases 13 points). It also helped to improve balance, lower body strength, flexibility and agility to a lesser extent. It is concluded that this type of intervention has positive effects for the sample in all the variables mentioned above.

## 1. Introduction

The ageing global population is increasing; in 2050, people over 65 will exceed the number of young people between 15 and 24 years old [1]. It is necessary to clarify that an older person is often defined as an adult up to 65 years of age; however, from a practical perspective, three age groups are distinguished post-65: those 65 to 74 years of age are referred to as ‘young older people’; those 75 to 84 are ‘older people’; and those >85 years are ‘older people [2]’.

Traditionally, there has been a negative image of the ageing process; an opinion that is erroneous, according to authors such as [3], who indicate that, during old age, many people maintain high physical and functional health. Therefore, having a positive perception of this stage allows us to take advantage of all the development possibilities presented by this group. This study concludes that, by engrossing themselves in active learning programmes at a later stage in life, older people can improve their own lifestyle and feel more capable of adapting to societal and social changes.

It is believed the ageing population share common objectives—for example, to extend the quality of life to older ages [4]. To achieve an adequate quality of life in the latter stages—that is, a satisfactory longevity—it is necessary that the person understands how to lead a balanced lifestyle, promoting health for greater autonomy and independence in their decisions and participating in activities that facilitate, among other things, correct self-care. Educational intervention work in these age groups is incredibly important [5].

Drawing from the ideas from previous programmes, a similar conclusion to other work [6] can be reached, who affirm that the inclusion of physical exercise programmes, adapted for older persons and taught by professionals within the realms of sports science and physical activity (PA), becomes a social need, understood as regular physical exercise. This provides a non-pharmacological strategy to age healthily and reduce the number of falls and thereby improve the health-related quality of life of older people.

Focusing on falls as a predictor of the loss of quality of life at an older age, there are works [7] that highlight that falls are a significant cause of death in the older population. An important field of work linking the consequences of falls and their proactive treatment is established [8]. The Ministry of Health, Social Services and Equality [9] state that, in most cases, the consequences of falls amongst the older population are very serious, resulting in more than 70% fractures or injuries and 10% with higher severity, leading to hospitalisations. Hospitalisations for falls are 5 times more frequent than for injuries due to other causes. Therefore, falls are a significant issue for an ageing society, meaning they are a concern at a general level, and their prevention is essential to minimize economic and social spending and for an individual to achieve a better quality of life [10]. In addition, it is pointed out that 50% of falls are repeated within the same year—that is, the initial fall itself increases the chances of recurrence. Therefore, in order to reduce the number of falls and increase the safety of a person considered fragile [11], it is necessary to focus physical exercise programmes and on ways to prevent falls and reducing their harmful consequences inside or outside the home [12]. In order to encourage greater participation in PA programmes, exercises must be achievable, accessible and varied [13].

It should be noted that tai chi, yoga and Pilates are all activities that encourage improvements in physical wellbeing, thus helping to reduce the risk of suffering a fall [14]; however, by reviewing the literature, it is evident there are even more specific programmes based on the proactive teaching of falls, such as the adapted utilitarian judo (JUA) [15] and Safe Fall [16]. The aforementioned programmes present a pioneering intervention, as they propose educational action strategies to teach how to fall safely, maintain mobility on the ground and return to standing [17]. In the same way, they have worked on aspects of falls for different populations, in order to achieve a measured way of teaching them. They have reached varied yet important conclusions, such as the fear of falling in older adults being reduced [18]; they highlight a possibility of reducing the severity of school-age injuries [19]. They propose developments of variables that are related to the improvement of quality of life [20] or improvements in factors studied, such as fear of falling or perception of health and physical condition during the COVID-19 pandemic, where the subject performed the JUA at home [21].

These programmes adapt traditional judo techniques in order to reduce impact when applied to the body [18]. They are based on the design of exercises, activities and games where the teaching of falls is introduced, focusing on the protection of the head, neck, hip and wrists [22]. 

These programmes are designed to incorporate exercises where strength, agility, basic and specific coordination and general and fine motor skills are worked on. These are easily transferable to the fundamental and essential activities of daily living, thus delaying higher levels of dependency and improving the quality of life in older adults [20]. Nonetheless, in addition to physical capacity and qualities, the combination of JUA, Safe Fall and Pilates offers specific work focused on the strength of the lower body and balance (static and dynamic), elements identified by different authors as determining factors in the prevention of falls in the older adult population [23,24]. 

From this prima in this research, the aim of this study is to determine and analyse the effects of a multidisciplinary intervention based on the Safe Fall, Safe Schools, adapted utilitarian judo (JUA) and Pilates programmes in a population of older people. 

## 2. Materials and Methods

### 2.1. Participants 

The sample was classified as healthy and pre-fragile within the fragility parameters for the older population [25]. The design was quasi-experimental, with pre- and post-measures of the control and experimental groups. They were selected using non-probabilistic-incidental sampling [26] for their accessibility (convenience sampling). Twenty-five subjects aged between 55 and 70 years took part in this study. The experimental group consisted of 13 women (63.85 ± 2.79 years). The anthropometric characteristics were adjusted to the average weight 69.15 ± 6.90 kg (BMI 26.59 ± 2.79) in women, and average height of 1.61 ± 0.05 m. The control group consisted of 12 women (63.08 ± 4.12 years). Their anthropometric characteristics were adjusted to average weight 64.00 ± 9.40 kg (BMI 25.05 ± 3.67), and average height of 1.60 ± 0.06 m. Regarding PA in the experimental group, 46% performed other PA in addition to Pilates—namely, hiking, yoga or maintenance gymnastics; the sample performed one or the other sporadically, as these were meetings held by the village council once a month with the aim of socialising rather than maintaining a level of physical activity. With respect to the control group, it is described that 50% performed other PA in addition to Pilates, including hiking, yoga or maintenance gymnastics sporadically.

The fall history of the sample included a fall in the previous 6 months. In the experimental group, 53.8% suffered a fall; of that percentage, 66.67% of these resulted in a fracture. In the control group, 33.3% suffered a fall; of that percentage, 25% had a fracture as a consequence.

As inclusion criteria, subjects had to be 55 years or over and with no diagnosed illnesses that would prevent them from exercising. Subjects were excluded if, for medical reasons, they had been advised against performing physical exercise; had suffered from congestive heart failure; felt chest pain, dizziness or angina during exercise; or had uncontrolled high blood pressure (160/100).

### 2.2. Instruments

Several instruments were used for the assessment of falls: the Quality of Life Questionnaire (SF-36) [27], relating to physical function, physical role, bodily pain, general health, vitality, social function, emotional role, mental health and health transition; the 8 Foot Up And Go test [28] to assess agility; the Chair Sit And Reach test [29], which assesses lower body flexibility; the Tinetti scale [30] for gait balance, which is an instrument used to determine the risk of falling in older persons; and the Chair Stand Test [31,32] to assess the lower body force. All instruments were applied before (pre-test) and after (post-test) the intervention.

As for the reliability of the tools used in the study, this was provided by an overall Cronbach’s alpha statistic value. It was noted that the Quality of Life Questionnaire (SF-36) was 0.91 (95% CI: 0.91–0.94); for the Tinetti scale, the statistic was 0.86 (95% CI 082–0.90); the 8 Foot Up And Go test was 0.98 (95% CI 0.96–0.99); Chair Sit And Reach test was 0.92 (95% CI 0.90–0.94); and, for the Chair Stand Test, it was 0.93 (95% CI 0.76–0.98).

The validity of the instruments was ascertained in two ways: content validity (a Content Validity Index [33] was obtained for all instruments above 00.95) and construct validity (this was carried out by a group of three experts of recognised standing in the field of physical activity for older adults).

### 2.3. Procedure

Both experimental and control groups received directed Pilates mat classes. For the experimental group (see Appendix A), specific sessions were designed that included exercises, activities and games of the proactive programmes of teaching falls, called JUA [15] and Safe Fall [16,19]. Programmes were implemented in an experimental group and control group over 12 weeks, with two weekly sessions of 60 min. The experimental group spent the first 10 min of the main part of the session carrying out Pilate exercises and the following 20 min doing JUA and Safe Fall exercises. These were related to working on movement, balance, coordination, basic motor skills, grip, flexibility, agility, strength exercises and exercises to work on falls, with self-loads, in sets of 10 repetitions and with active rest of 1 min. Heart rates were monitored with the wearable Garming Forerunner 245 (range between 90–160 bpm) and were collected using the Garmin Connect™ Challenges App. The researchers provided one unit of this wearable device to each study participant before each session. The exercises proposed focused on giving overall treatment to the prevention of falls and their consequences, with a utilitarian function for the group studied. The choice of exercises selected was based on a programme of increasing difficulty, following the principles of the programmes, assimilation and, above all, participant safety (Figure 1). They introduced activities and games belonging to the Safe Fall programme, giving dynamism to the intervention, thus achieving adherence to the activity.

Control group subjects were applied an intervention programme based on classes directions of Pilates (light and moderate intensity, aerobic activity, muscle-strengthening activity, coordination, balance, and flexibility), over the course of 12 weeks, with two weekly sessions each lasting 60 min in duration, in sets of 10 repetitions and with active rest of 1 min. Comparable to the experimental group, heart rates were monitored with the wearable Garming Forerunner 245 (range between 90–160 bpm) and were collected using the Garmin Connect™ Challenges App. The researchers provided one unit of this wearable device to each study participant before each session.

All sessions in both groups (experimental and control) were developed by the same trained and experienced instructor. All subjects in the sample were informed about the objectives of the study and agreed to participate after giving informed consent. Additionally, the Ethics Committee of the Biomedical Research of Andalusia approved the waiver of consent for this study (Códigos; JUA: 1148-n-18/Safe Fall; 0021-m1–18).

### 2.4. Data Analysis

A descriptive statistical analysis was carried out using the statistical programme SPSS (V.26.0.), with the category variables presented with the mean and standard deviation. The Student’s t-test was utilised for independent samples, in order to analyze the significance in the mean difference. In all cases, the probability *p*-value ≤ 0.05 was considered to be significant.

## 3. Results

Regarding the variable quality of life, the results show that the experimental group increased its score in all dimensions; the highest values were found in the social function of this group, with 94.81 points (approaching a maximum score of 100), 2 points above the results obtained in the same dimension of the control group.

It should be noted that two other dimensions of the experimental group stand out, due to the important difference between the values obtained in the pre-test and post-test. Specifically, body pain (pre-test: 65.19 and post-test: 77.69) and health transition (pre-test: 50.00 and post-test: 63.46). 

Finally, in relation to the control group, it should be noted that their score decreased in the dimensions of physical function (pre-test: 87.50 and post-test: 86.67) and general health (pre-test: 62.50 and post-test: 61.66). 

Next, referring to the variable of balance, after putting into practice the Tinetti test (evaluation gait and balance), it is concluded that all subjects obtained the maximum score in practice. As a consequence, the variations that were found in the measurements, once the intervention was carried out, will depend only on the fluctuations in the equilibrium. It is evident that the changes are not especially significant: control group (pre-test: 26.67 and post-test: 27.00) and experimental group (pre-test: 26.54 and post-test: 27.23). However, an analysis of the results in this test for each of the participants indicates that 8.33% of the control group had an average risk of falling, and the remaining 91.67% had a low risk. In the experimental group, 38.46% had a medium risk of falling, and the remaining 61.54% had a low risk. After the intervention, 100% of participants had a low risk of falling. 

In relation to the variable of lower body strength, the results illustrate improvements in both groups, with a greater increase in the experimental group: control group (pre-test: 13.92 and post-test: 15.58) and experimental group (pre-test: 13.46 and post-test: 17.38). Upon further analysis, the results for each of the participants in the Chair Stand Test revealed that 8.33% of the participants from the control group and 15.38% of the participants from the experimental group were below the values considered normal. For the scores to be considered normal, they must be between 12 and 17 (60–64 years) and between 11 and 16 (65–69 years). At the end of the intervention, all subjects met the established values. 

The results in the flexibility variable demonstrate notable improvements in both groups: control group (pre-test: −0.56 and post-test: +3.07) and experimental (pre-test: −0.36 and post-test: +3.85). Furthermore, the results for each of the participants in the Chair Sit And Reach test show that 58.33% of the control group and 46.15% of the experimental group did not comply with the values considered normal: −0.5 and −5.0 (60–64 years) and −0.5 and +4.5 (60–64 years), respectively. After the intervention, 33.33% of the experimental group and 15.38% of the control group did not meet the values. 

Finally, with regard to the agility variable, it was observed that there was a significant difference in the post-test results of the experimental group, with a *p* = 0.007. In addition, the results of the 8 Foot Up And Go Test indicate that, in the pre-test, 16.67% of the participants in the control group and 46.15% of the experimental group were not within the values considered normal; for the scores to be considered normal, they have to be between 6.0 and 4.4 (60–64 years) and 6.4 and 4.8 (65–69 years). At the end of the intervention, all participants, except 8.33% of the control group and 7.8% of the experimental group, met these values. 

Table 1 illustrates the results of the pre- and post-test of the experimental and control groups for each of the variables.

## 4. Discussion

After analysing the results, it has been shown that the experimental group improved its score in all variables (quality of life, balance, lower body strength, flexibility and agility). The benefits the subjects obtained, as a result of this improvement, allowed this group to sit and physically get up, since they increased their lower body strength levels. It was noted that activities were applied in which the older persons were taught to return to a standing position. This is assumed to improve their quality of life because, in helping them to perform activities from their daily routine, such as playing on the floor with grandchildren, these physical activities would enhance future opportunities. This coincides with [3], where they stress the importance of generativity as a factor that increases the quality of life of the ageing population.

The SF-36 questionnaire results affirm that the intervention, introducing JUA and Safe Fall exercises, activities and games, improves all dimensions of quality of life. Some findings recorded incredibly high scores—for example, physical function (84.23); social function (94.81); and emotional role (89.74). Comparing the results with other studies [18,19], they indicate that a proactive programme that includes the teaching of falls can improve the quality of life for at-risk populations.

That being said, this study has not carried out the JUA and Safe Fall programmes in their entireties; however, results obtained exhibit high levels of similarity to existing studies, reinforcing the data collected. The results collected for the rest of the variables (balance, lower body strength, flexibility and agility) show that this study’s intervention had a very positive effect, since the score of all the variables in the post-test was higher than in the pre-test. This is evident in some studies [18,19,20] that affirm that the JUA and Safe Fall programmes have multiple benefits on dynamic equilibrium, strength and agility, managing to reduce falls and thus potentially improving the quality of life. Positive connections to other work [31,32] can be made, despite performing a variation on the form of intervention. This legitimises that physical exercise that combines strength, balance and flexibility delay dependence and improve the quality of life of older adults [34]. 

Some work [35] point out that exercise programmes that combine strength, balance, agility and flexibility, with an average duration of 12 weeks, improve quality of life, which reflects the findings of this study. 

The results obtained in this study coincide with others that affirm Pilates improves balance and reduces the risk of falling [36]. It improves the psychological variables of the SF-36 questionnaire [37] and reduces cognitive impairment [38]. Taking into account these data, we understand the improvements found in the control group of this study; however, this type of therapy does not contemplate the actions that can be carried out when a fall occurs, which is where the JUA and Safe Fall programmes represent a unique intervention [19].

In short, PA improves the quality of life of older adults [39] and, combined with programmes based on proactive fall education, such as JUA and Safe Fall, older adults will also benefit from specific physical exercise, such as falls, where they can continue to learn and improve their quality of life [20]. 

## 5. Conclusions

The main conclusions of this study are that the intervention improves all dimensions of the SF-36 questionnaire and, therefore, helps to improve the quality of life for the participants observed. It indicates that all subjects within the study had a lower risk of falling and showed increased positive effects on the strength of the lower body, improving up to 5 points above the pre-test score. Significant developments were obtained on the flexibility variable, improving the pre-test score by 3 points and presenting substantial improvements in the agility variable. 

Therefore, we conclude that interventions based on the Pilates method, introducing exercises, activities and games of the JUA and Safe Fall programmes, are beneficial for the improvement of the quality of life for the ageing population. 

Being aware of the various benefits that PA brings to the ageing population, it is necessary to promote the practice of physical exercise taught by professionals in the field of PA. In addition, the effectiveness of our intervention on the participants of this study makes it of special interest, in that we use exercises from these programmes in our sessions to teach older people how to fall, get up and recover mobility on the ground after the fall, thus improving their quality of life or reducing the potential for harm caused from a fall. 

## Figures and Tables

**Figure 1 ijerph-20-01246-f001:**
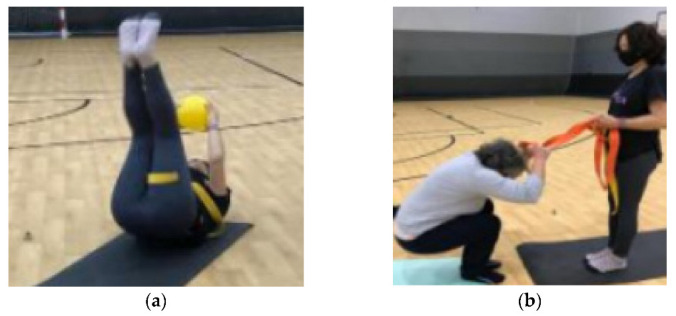
(**a**) Fall training with implements; (**b**) assisted falling exercises.

**Table 1 ijerph-20-01246-t001:** Pre- and post-test results of the control and experimental groups.

		Control Group					Experimental Group				
		Pre-Test		Post-Test			Pre-Test		Post-Test		
		M	SD	M	SD	Sig.	M	SD	M	SD	Sig.
Quality of life	Physical function	87.50	15.00	86.67	11.93	0.219	79.61	16.13	84.23	13.67	0.115
	Physical role	72.92	44.54	83.33	38.92	0.445	84.61	29.82	86.53	28.17	0.687
	Bodily Pain	74.17	21.65	85.83	21.59	0.333	65.19	23.55	77.69	21.64	0.078
	General Health	62.50	16.99	61.66	16.00	0.678	62.69	17.63	67.69	16.91	0.103
	Vitality	63.75	16.80	66.66	15.57	0.666	60.38	21.16	68.84	13.87	0.550
	Social function	88.33	17.94	92.92	11.52	0.619	84.04	23.95	94.81	10.68	0.320
	Emotional role	69.44	45.96	83.33	38.92	0.446	82.05	35.01	89.74	25.05	0.221
	Mental health	70.00	15.49	76.00	15.15	0.894	68.92	23.28	76.92	16.26	0.105
	Health transition	41.66	12.31	52.08	12.87	0.532	50.00	17.68	63.46	21.93	0.430
	Balance	26.67	1.37	27.00	1.13	0.847	26.54	1.85	27.23	1.09	0.221
	Lower body force	13.92	2.15	15.58	2.07	0.615	13.46	2.30	17.38	2.76	0.345
	Flexibility	−0.56	2.12	3.07	3.09	0.593	−0.36	2.07	3.85	2.12	0.100
	Agility	5.32	0.73	5.04	0.70	0.440	6.13	0.64	5.25	0.63	0.007

Note: M = mean; SD = standard deviation; *p*≤ 0.05. Source: authors.

## Data Availability

Not applicable.

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
