# Peer review of "Effects of the Implementation of an Intervention Based on Falls Education Programmes on an Older Adult Population Practising Pilates–A Pilot Study"

_ijerph, 2023, doi:10.3390/ijerph20021246_

Round 1

Reviewer 1 Report

I have attached the review.

Author Response

Answer to the Reviewer’s comments

Manuscript ID: ijerph-2035300

Dear Editor

Enclosed you will find a revision of our manuscript, Effects of the implementation of an intervention based on falls education programs on an older adult population. A pilot study. We would like to thank the Editor for giving us the opportunity to resubmit our work and to the reviewers for their thoughtful and constructive comments. We have considered all the suggestions and have incorporated them into the revised manuscript. An itemized point-by-point response to the reviewer’s comments is presented below. 

Comments to the Author

Dear reviewer:

Thank you again for your comments.

[Introduction session]

  1. The risks of aging and falls are well understood through the introduction. However, the need for fall prevention effects of the JUA, safe-fall and Pliates involved in this study is not highlighted

Response 1: Page 2, lines 88-92. The following information and reference is added to the text.

[Meterials and Methods session]

  1. Participants session of line 102: This study was understood as a study to verifty the effectiveness of intervention program. However, the exercise rate of the experimental and control group was high. Why did this study not recruit non-exercise subjects? Also, does marital status affect outcome variables?

Response 2: No sedentary subjects were chosen for this research as the sample selection was non-probabilistic-incidental or convenience (Salking, 1999).

As for marital status, this was only intended to be descriptive. The reviewer's question has made us reflect on the relevance of this descriptive in the manuscript and, therefore, as it is not significant, the authors chose to eliminate this descriptive.

Page 3, lines 108-114, the paragraph is modified.

  1. Participants session of line 107: This study to verify the effectiveness of fall prevention, but there seems to be a difference in the experience of falling between the two groups. Have additional controls been implemented so that these differences do not affect outcome variables?

Response 3: The reviewer's assessment is correct. We would like to point out that this data is not significant for our study, as the falls in both groups (experimental and control) occurred more than a year before the implementation of the programme and none of them had functional limitations linked to their falls. Therefore, it was considered that there was no need for any type of addictive control.

  1. Participants session of line 107: Similar to No. 3, the experimental group experienced more fractures due to falls than the control group. If so, it cannot be overlooked that the results were improved by fracture recovery, not by interventional exercise effects.

Response 4: Similar to Nº 3. We add that in all cases the participants in the study stated that they had recovered from the injuries caused by their fall, for which purpose, prior to the intervention in the programme, they all provided an official medical certificate indicating that they were not limited in their physical activity.

In this sense, the authors consider that the improvement obtained by the subjects after the intervention can exclude recovery from the injury. 

  1. Instruments session of line 117: The evaluation criteria suh as SF-36 and Tinetti scale used in this study were not clearly explained. Because of this, the interpretation of the results is ambiguous, so it is necessary to dexcribe the description of the evaluation tool in detail.

Response 5: Thank you very much for your appreciation.

In the section on instruments, each of the instruments used is named with its reference, the reader of the article interested in the specific protocol of a test, if desired, can find the protocol described in detail in that reference. Specifically, in this intervention, we have followed the protocol described in detail in each of the references indicated.

We have chosen to reference each instrument and not to explain in detail each of the tests in the instrument section due to the limited space available in the number of words in the article. 

However, if the reviewer and the editor deem it appropriate, we have no objection to expanding this section, either within the article or as an appendix.

  1. Instruments session of line 117: The reliability and validity of the evaluation tool used should be presented.

Response 6. We agree with the reviewer. We therefore add the validity and reliability of the tools used. P. 3, lines 133-142.

  1. Procedure session of line 126: The composition of the exercise program is difficult to understand. It needs to be presented in table format.

Response 7. Due to the length of the document, we have tried to respond to the reviewer by adding the composition of the exercises in the text: pages 3-4, lines 148-153.

  1. Procedure session of line 140: Were the intensity of the exercise applied to the two groups equally controlled? Further explanation of the exact exercise intensity is needed.

Response 8. Information on the type and intensity of exercises is expanded. Confer. Pag. 3, lines 148-153 and pag. 4, line 165-166.

  1. Data Analysis session of line 154: Statistical significance must be described in terms of an exact p value for the probability of significance other than +≤ 0.05.

Response 9. Table 1 shows the significance data for all variables in the control and experimental groups.

[Results session]

  1. In Table 1, a significant difference was found only in “agility”. However, the overall interpretation of the results is dexcribed as indicating a significant difference. It is only necessary to describe the statistical significance in the results session.

Response 11. Done.

  1. Resuls session of line 180: It is necessary to clearly explain how “lower body strength” is calculated in the [Materials and Methods] session.

Response 12. As in point 5 of this review, the authors consider that, by naming each of the instruments used with its reference, the reader of the article interested in the specific protocol of a test can, if desired, find the protocol described in detail in that reference. Specifically, in this intervention, we have followed the protocol described in detail in each of the references indicated.

We have chosen to reference each instrument and not to explain in detail each of the tests in the instrument section due to the limited space available in the word count of the article. 

However, if the reviewer and editor deem it appropriate, we have no objection to expanding this section, either within the article or as an appendix.

  1. Resuls session of line 174, 191 and 196: How was the percentage value described in that line calculated? In addition, if these results important, it is necessary to present them in the results table and suggest statistical significance.

Response 13. We appreciate the reviewer's comment on this aspect. We would like to clarify, here, that these data are not percentage values. As explained in the description of the instrument, by its original authors, reference (27) the dimensions of the SF-36 the higher the score, the more favourable the health status, with 100 being the maximum score and 0 the minimum.

We have also added the significance of these variables in Table 1.

  1. Table 1 in the results part(line 208): The evaluation tool used in [Instrument] and the description of the variables presented in the result table are different. Need to match.

Response 14. The interpretation of the nomenclature used in the [instrument] section and in table 1 is clarified by matching it. Page 3, lines 127-131.

  1. Table 1 in the results part(line 208): Correct the abbreviation of “Standard deviation” to “SD” in the table

Response 15. Many thanks to the reviewer for his correction. Done

  1. Table 1 in the results part(line 208): Is “Flexibility” not pre-test? If you have done a pre-test, please describe it.

Response 16. Many thanks for the reviewer's comment. We have detected a transcription error. The flexibility pretest data have been added to table 1.

[Discussion session]

  1. Discussion session of line 220: It was judged that it would be helpful to read the discussion in the order presented in the result table.

Response 17. The authors agree with the reviewer. The text is restructured.

  1. Discussion session of line 243: Unlike Pilates in this study, it is understood to suggest improvements in JUA and safe-fall programs. However, the difference between the two exercise interventions was not statistically confirmed in the results of this study. Therefore, it is necessary to rewrite this part based on the results.

Response 18. Done. The text is rewritten based on the results as advised by the reviewer.

Reviewer 2 Report

Dear Authors

You have written an interesting study, however, some parts need to be addressed for greater clarity.

In the title, there is no mention of Pilates. From the first instance, the title suggests that only safe falls are the object of this research. Amend

Abstract: report p values of significant findings.

Keywords should not be from the title.

The introduction is nicely written and clearly leads to the main rationale of the study. However, I am missing a clear definition of an elderly person from an age perspective as this is then the basis for your sampling. Amend and back up with references

Methods: how was your sample size determined (G*Power or any other method). Report

Participants - first you write that the  5 subjects aged between 55 and 70 took part in this study, but later you add that the inclusion criteria were for over 60 years old participants. What was it then? Amend

How were they divided into experimental and control groups? Report

Prior physical activity levels need to be better defined and reported as they can significantly affect your results.

Line 115 -- how was the blood pressure monitored? report

The programs of the experimental group and pilates program need to be presented in detail. In the current description, the study is not reproducible. Which exercises, repetitions, sets, break, etc.

Additionally, all the tests under the Instruments paragraph need to be explained how were they executed, the accessories used, repetitions, and what result was taken into further analysis.

Overall a really poorly described methods section which, in its current state, makes this case study irreducible.

Therefore I a rejecting this paper.

I hope that the feedback will help the authors to improve and resubmit their study.

Kind regards

Author Response

Answer to the Reviewer’s comments

Manuscript ID: ijerph-2035300

Dear Editor

Enclosed you will find a revision of our manuscript, Effects of the implementation of an intervention based on falls education programs on an older adult population. A pilot study. We would like to thank the Editor for giving us the opportunity to resubmit our work and to the reviewers for their thoughtful and constructive comments. We have considered all the suggestions and have incorporated them into the revised manuscript. An itemized point-by-point response to the reviewer’s comments is presented below. 

Comments to the Author

Point 1: In the title, there is no mention of Pilates. From the first instance, the title suggests that only safe falls are the object of this research. Amend

Response 1: Dear reviewer:

Thank you again for your comments.

We introduced your indications: “Effects of the implementation of an intervention based on falls education programs on an older adult population practising pilates. A pilot study”

Point 2: Abstract: report p values of significant findings.

Response 2: Thank you very much for your input, it has been done. Added to document, p. 1, lines 20

Point 3: Keywords should not be from the title.

Response 3: Dear reviewer:

Thank you again for your comments.

We introduced your indications: “Adapted Utilitarian Judo; Safe Fall-Safe Schools; Pilates method; quality of life” Added to document, p. 1, lines 25

Point 4:The introduction is nicely written and clearly leads to the main rationale of the study. However, I am missing a clear definition of an elderly person from an age perspective as this is then the basis for your sampling. Amend and back up with references

Response 4: Thank you very much for your review. After a thorough search and reading of the scientific literature, we see the importance of introducing the following clarification. In order to establish a clear definition of an older person from the point of view of age, we base ourselves on the following author García Peris, P (2004), who considers an adult person up to the age of 65 years. After this age, it is accepted as the definition of an older person. From a practical point of view, three age groups are distinguished; from 65 to 74 years we speak of young old people, from 75 to 84 of old people and above 85 years of age of old people.

Added to document, p. 1, lines 27-31

Point 5: Methods: how was your sample size determined (G*Power or any other method). Report

Response 5: The reviewer is advised that how the sample size was determined can be found in the method. We provide the paragraph here: “They were selected using non-probabilistic-incidental sampling [23] for their accessibility (convenience sampling)” Added to document, p. 3, lines 102-103

Point 6: Participants - first you write that the 5 subjects aged between 55 and 70 took part in this study, but later you add that the inclusion criteria were for over 60 years old participants. What was it then? Amend

Response 6: Thank you very much for your appreciation, of course it is a mistake that we assume and thanks to your review improves our article, we will change it in the paragraph as follows: “As inclusion criteria, subjects had to be 55 or over and with no diagnosed illnesses that would prevent them from exercising” Added to document, p. 3, lines 120

Point 7: How were they divided into experimental and control groups? Report

Response 7: Thank you for your appreciation, your comment really improves the description of the sample. The division into experimental and control group was made from the same group of women who were doing pilates, the division was random. The subjects were numbered randomly and anonymously, the even numbered subjects went to the control group and the odd numbered subjects went to the experimental group.

Point 8: Prior physical activity levels need to be better defined and reported as they can significantly affect your results.

Response 8: Thank you for your comment, as we agree with you that this may be an aspect that could affect the results of our study. In the section on the description of the sample, we can read that the experimental group, specifically 46% did other PA in addition to Pilates, namely hiking, yoga or maintenance gymnastics, the sample did one or the other on an exporadic basis, as they were meetings held by the village council once a month with the aim of socialising rather than maintaining a level of physical activity.

With respect to the control group, it is described that 50% did other PA in addition to Pilates, this being hiking. There is an error here, as it is the same activity as hiking, yoga or maintenance gymnastics offered on an exporadic basis.

Added to document, p. 3, lines 109-115

Point 9: Line 115 -- how was the blood pressure monitored? report

Response 9: Thank you for your suggestion. We have established participant inclusion criteria within our study design as this is important to consider in all high quality research protocols. Within these we have included clinical characteristics and controlled for these by providing an official medical certificate of the subjects at the start of the intervention.

Point 10: The programs of the experimental group and pilates program need to be presented in detail. In the current description, the study is not reproducible. Which exercises, repetitions, sets, break, etc.

Response 10: Dear reviewer, thank you for your comment. We have details of each of the sessions that took place in the control group and in the experimental group. This material takes up a lot of space and it is impossible for us to provide it, but we will add this text to the article to help, together with the bibliographic references that we provide, where the whole programme is developed, to be able to replicate it: "These exercises in the experimental group consisted of working on movement, balance, coordination, basic motor skills, grip, flexibility, agility, strength exercises and exercises to work on falls, with self-loads, in series of 10 repetitions and with active rest of 1 minute". Added to document, p. 3-4, lines 149-154.

Point 11: Additionally, all the tests under the Instruments paragraph need to be explained how were they executed, the accessories used, repetitions, and what result was taken into further analysis.

Response 11: Thank you very much for your appreciation.

In the section on instruments, each of the instruments used is named with its reference, the reader of the article interested in the specific protocol of a test, if desired, can find the protocol described in detail in that reference. Specifically, in this intervention, we have followed the protocol described in detail in each of the references indicated. However, following their indications, the appropriate explanation has been added. Added to document, p. 3, lines 127-143.

Round 2

Reviewer 1 Report

Thank you for your correction to the review comment.

Overall, the comments on the review were well structured.

Author Response

Many thanks to the reviewer for his work. It has greatly contributed to the qualitative improvement of the document.
Sincerely,
The authors

Reviewer 2 Report

Dear Authors,

Thank you for addressing the majority of my questions. On some, we have to agree to disagree. Overall the manuscript quality improved.

However, taking a lot of space excuses in MDPI journals is a poor one. I would suggest adding the exact training programme as an appendix to ensure the repeatability of your study. Without it, no one can exactly replicate your study and check your findings. I hope this is clear.

There are varous kinds of pilates. Therefore, write down the order of exercises and also was there a progresive overload priciple applied. If yes report the change. 

Line 154 - how was the heart rate monitored? report

Kind regards

Author Response

Answer to the Reviewer’s comments

Manuscript ID: ijerph-2035300

Dear Editor

Enclosed you will find a revision of our manuscript, Effects of the implementation of an intervention based on falls education programs on an older adult population. A pilot study. We would like to thank the Editor for giving us the opportunity to resubmit our work and to the reviewers for their thoughtful and constructive comments. We have considered all the suggestions and have incorporated them into the revised manuscript. An itemized point-by-point response to the reviewer’s comments is presented below. 

Many thanks to the reviewer for his work. It has greatly contributed to the qualitative improvement of the document.
Sincerely,
The authors

Comments to the Author for the reviewer

 Point 1: I would suggest adding the exact training programme as an appendix to ensure the repeatability of your study. Without it, no one can exactly replicate your study and check your findings.

Response 1: We agree with the reviewer. See appendix added on page 3, Line 146 and the appendix with the training programme is added to the document.

Point 2: There are varous kinds of pilates. Therefore, write down the order of exercises and also was there a progresive overload priciple applied. If yes report the change.

Response 2: For this intervention we have used the Pilates Mat (this information is added on page 3, line 145). In this work, the principle of progressive overload has not been applied, as the authors consider that the correct execution of the exercises is more important than the number of times they are performed and their overload. Pilates exercises are marked with a maximum number of repetitions but, if done correctly, their effectiveness makes it unnecessary to increase the number of times they are repeated.

The order of the exercises can be found in the appendix.

Point 3: Line 154 - how was the heart rate monitored? report

Response 3: pag. 4, lines 154-157 for experimental group and lines 167-170 for control group.

The authors have detected and corrected an error in the minimum bpm, which should be 90 bpm.
